# Mesenchymal Stem Cell Therapy for Bone Repair of Human Hip Osteonecrosis with Bilateral Match-Control Evaluation: Impact of Tissue Source, Cell Count, Disease Stage, and Volume Size on 908 Hips

**DOI:** 10.3390/cells13090776

**Published:** 2024-05-01

**Authors:** Philippe Hernigou, Yasuhiro Homma, Jacques Hernigou, Charles Henri Flouzat Lachaniette, Helène Rouard, Sophie Verrier

**Affiliations:** 1Orthopedic Department, University Paris East, Hopital Henri Mondor, 94000 Creteil, France; charles-henri.flouzat-lachaniette@aphp.fr; 2Department of Orthopaedics, Faculty of Medicine, Juntendo University, Bunkyo-ku, Tokyo 113-8421, Japan; yhomma@juntendo.ac.jp; 3Department of Orthopaedic Surgery and Traumatology, EpiCURA Baudour Hornu Ath Hospital, 7331 Hainaut, Belgium; jacques.hernigou@gmail.com; 4Établissement Français du Sang, University Paris East, 94000 Creteil, France; helene.rouard@efs.sante.fr; 5AO Research Institute Davos, Clavadelerstrasse 8, 7270 Davos, Switzerland; sophie.verrier@aofoundation.org

**Keywords:** hip osteonecrosis, stem cells, mesenchymal stem cells, osteonecrosis repair, bone marrow autologous concentrate cells, allogenic expanded stem cells, autologous expanded stem cells, core decompression, bone repair

## Abstract

We investigated the impact of mesenchymal stem cell (MSC) therapy on treating bilateral human hip osteonecrosis, analyzing 908 cases. This study assesses factors such as tissue source and cell count, comparing core decompression with various cell therapies. This research emphasizes bone repair according to pre-treatment conditions and the specificities of cell therapy in osteonecrosis repair, indicating a potential for improved bone repair strategies in hips without femoral head collapse. This study utilized a single-center retrospective analysis to investigate the efficacy of cellular approaches in the bone repair of osteonecrosis. It examined the impact on bone repair of tissue source (autologous bone marrow concentrate, allogeneic expanded, autologous expanded), cell quantity (from none in core decompression alone to millions in cell therapy), and osteonecrosis stage and volume. Excluding hips with femoral head collapse, it focused on patients who had bilateral hip osteonecrosis, both pre-operative and post-operative MRIs, and a follow-up of over five years. The analysis divided these patients into seven groups based on match control treatment variations in bilateral hip osteonecrosis, primarily investigating the outcomes between core decompression, washing effect, and different tissue sources of MSCs. Younger patients (<30 years) demonstrated significantly better repair volumes, particularly in stage II lesions, than older counterparts. Additionally, bone repair volume increased with the number of implanted MSCs up to 1,000,000, beyond which no additional benefits were observed. No significant difference was observed in repair outcomes between different sources of MSCs (BMAC, allogenic, or expanded cells). The study also highlighted that a ‘washing effect’ was beneficial, particularly for larger-volume osteonecrosis when combined with core decompression. Partial bone repair was the more frequent event observed, while total bone repair of osteonecrosis was rare. The volume and stage of osteonecrosis, alongside the number of injected cells, significantly affected treatment outcomes. In summary, this study provides comprehensive insights into the effectiveness and variables influencing the use of mesenchymal stem cells in treating human hip osteonecrosis. It emphasizes the potential of cell therapy while acknowledging the complexity and variability of results based on factors such as age, cell count, and disease stage.

## 1. Introduction

Osteonecrosis of the femoral head (ONFH) is a condition in which a part of the femoral head becomes necrotic, associated with a disruption of the blood supply or due to some diseases. The necrosis usually involves the weight-bearing upper part of the femoral head. As the disease progresses, the mechanical strength of the necrotic bone is reduced, leading to eventual collapse. There are sufficient data to support such a pathophysiologic cascade: marrow and bone necrosis; subchondral fracture; and collapse of the femoral head (Figure 1A). This evolution is graded (Figure 1B) in different stages [1] and leads to subsequent secondary arthritis of the hip and to total hip arthroplasty in most of the cases where natural evolution is observed [2].

Non-traumatic ONFH typically affects adults under 50 years old, which frequently leads to the collapse of the femoral head. An estimated 10,000 new people [3] are thought to be impacted by the disease each year in the US. In Japan [4], the yearly incidence rate was 1.91 per 100,000 people, and the predicted annual incidence for 2015 was above 2400. The estimated prevalence of ONFH rose from 9870 in 2002 to 18,691 in the Republic of Korea [5]. A recent large-scale epidemiological survey conducted in China [6] found that 8.12 million Chinese people overall had non-traumatic ONFH. Given the prevalence of sickle cell illness in Brazil, India, and Africa [7,8,9], the total number of sickle cell patients who have epiphyseal necrosis is most likely several tens of millions. Thus, the global total of individuals suffering from non-traumatic hip osteonecrosis (ON) may be as high as 20 million.

In addition, the use of corticosteroids [4], excessive alcohol consumption, excessive tobacco use, systemic lupus erythematosus, organ transplantation, infection with the human immunodeficiency virus, sickle cell disease, coagulopathies, multiple genetic factors, Caisson disease (a disease of deep-sea divers), myeloproliferative diseases (diseases of the bone marrow packing), and radiation necrosis are known risk factors for the developments of non-traumatic ONFH.

Hip osteonecrosis has a limited capacity to regenerate, leading to collapse and joint deterioration and, consequently, hip arthroplasty [10]. In addition to conservative measures and traditional surgical approaches, such as core decompression alone [11] or osteotomies [12], mesenchymal stem cells (MSCs) and cell therapy have also been studied in the management of osteonecrosis repair in recent decades [13]. The corresponding author’s experience [14] of over three decades has shown that tissue repair of osteonecrosis evaluated with MRI is variable in quality and depends not only on the cause and stage of osteonecrosis [15] but also on the cell-based treatment applied, i.e., cell number or source.

Although pharmacokinetics generally displays a dose-dependent response to a medicament, such a response does not always appear consistent for cell therapy, probably because stem-cell-assisted tissue healing processes require intricate biological mechanisms and dynamic interactions between various cell types. The variability of cells, growth conditions, administration modalities, and recipient conditions—all of which vary significantly in ongoing clinical trials and lab research—complicates MSC therapy further. For example, in osteonecrosis, many variables are recognized as factors accelerating collapse in osteonecrosis, and these factors may also act as limitations for repairing tissue with cell therapy. How these variables may positively or negatively affect the repair is probably not identical for each patient, according to the cause of osteonecrosis, the stage, the size, and probably many other factors [15,16].

Ischemic osteonecrosis also leaves extensive necrotic cell and fat debris in the marrow space, which is a cause of inflammation [17,18]. While acute inflammation is required for bone repair, chronic inflammation leads to further tissue damage and fibrosis. Clearance of necrotic cells and fat debris in the marrow space by macrophages [19] is obtained through phagocytosis. This means that other cells than pure MSCs are probably involved for the repair process.

Given the number of factors that can influence the repair of osteonecrosis, this study was limited to investigating only few factors of them, i.e., the number of cells, the quality of the cells, the volume of the injection, the washing effect of the liquid injection with the cells, and the core decompression effect, to avoid a certain number of biases. Only patients who did not show any post-operative treatment failure (hip collapse) after treatment were included, to allow an accurate evaluation of repair on MRI, an evaluation which is usually difficult in the presence of collapse. Patients with bilateral symptomatic hip necrosis with a similar volume of osteonecrosis were selected. Each side was treated differently during the same anesthesia. Therefore, this retrospective study evaluated factors that best determine the repair of osteonecrosis in the absence of collapse. This is not a study to research the best technique to prevent collapse, but it is a study to examine what factors influence repair in hips without collapse and the benefit of MSCs therein. The authors hypothesized that higher cell counts and specific tissue sources would result in significantly improved bone repair outcomes in early stage hip osteonecrosis compared to lower cell counts and different tissue sources.

## 2. Methods

### 2.1. Patients Selection

This single-center retrospective analysis aimed to assess the impact of various factors on osteonecrosis repair using a cellular approach: the effect of the tissue source (autologous concentrate, allogenic, expanded); the influence of cell quantity, ranging from no cells (core decompression alone) to millions of cells, and the cell density relative to the injection volume; and, finally, the influence of osteonecrosis stage, volume, and cause. This study also encompassed a comparison with the natural progression of hip osteonecrosis.

This retrospective study excluded hips with femoral head collapse. It specifically focused on patients who had both pre-operative and post-operative MRIs with a follow-up of more than 5 years. Out of a total of 4305 patients with 7549 cases of hip osteonecrosis, after exclusion as defined in Figure 2, seven groups were identified (total number of patients = 454, total number of ONFH = 908). These seven groups consisted of patients with bilateral symptomatic hip osteonecrosis who received different treatments on each side. Some groups had many patients, while others had a tiny population. Only patients with a similar pre-operative volume of osteonecrosis (less than a 5 cc—cubic centimeters—volume difference) on both sides were included. Hip osteonecroses classified as stage I and stage II were considered, while other stages were not included in the analysis.

Group 1 compared core decompression (CD) versus natural evolution (NE) in 52 patients with surgery performed between 1985 and 1988; 12 patients had bilateral stage I, 14 had bilateral stage II, and 26 patients had stage I on one side (12 with CD, 13 with NE) and stage II on the other (13 with CD, 12 with NE).

Group 2 compared core decompression versus the washing effect (WE) of 20 cc; 50 patients had surgery performed between 1985 and 1988. At the time when the corresponding author introduced the technique of injecting fluid in the osteonecrosis, the idea was not to wash the osteonecrosis. The rationale was to inject a contrast material to study the venous return of the osteonecrosis and a diluted anesthetic solution to judge the post-operative analgesia compared to the opposite side without a local anesthesia solution. Ten patients had bilateral stage I, 15 had bilateral stage II, and 25 had stage I on one side (11 with CD, 14 with WE) and stage II on the other (14 with CD, 11 with WE).

Group 3 had patients with core decompression alone versus bone marrow autologous concentration (BMAC) injection of 20 cc of cell therapy. Twenty mL of BMAC were prepared through centrifugation of around 100 mL of autologous bone marrow aspirated from the lilac crests. In the group 3, 234 patients had surgery performed between 1992 and 2000. Sixty-eight patients had bilateral stage I, 74 had bilateral stage II, and 92 patients had stage I on one side (41 with CD) and stage II on the other (37 with CD). These patients are a part of previous studies [15,20].

Group 4 compared autologous BMAC cell therapy (20 cc) versus washing (WE) without cells (20 cc): 25 patients had surgery between 1990 and 1992. Four patients had bilateral stage I, seven had bilateral stage II, and 14 had stage I on one side (six with BMAC) and stage II on the other (eight with BMAC).

Group 5 compares autologous BMAC cell therapy versus allogenic cell therapy (Allo CT). These 38 cases included osteonecrosis in children, adolescents, and young-adult leukemia survivors. Due to the low number of stem cells [21] in the bone marrow, these patients had sequential surgery between 1990 and 2015. They had bilateral necrosis treated on one side with autologous concentrated marrow autograft (BMAC) and, on the other, with allogeneic stem cells (Allo CT). Eight patients had bilateral stage I, ten had bilateral stage II, and 20 had stage I on one side (eight with BMAC) and stage II on the other (12 treated with CD).

Group 6 compares autologous BMAC cell therapy (20 cc) versus autologous expanded (Auto Exp CT) stem cells (7 cc). These seven patients were treated on one side with autologous, expanded, bone marrow-derived mesenchymal stromal cells in a multicentric trial with a minimum of five years’ follow-up [22]. Three patients had bilateral stage I, one had bilateral stage II, and three had stage I on one side (1 with BMAC) and stage II on the other (2 with BMAC).

The Group 7 autologous BMAC (20 cc) versus BMAC (20 cc) + WE of 20 cc performed before injecting BMAC were compared. This study on these 48 patients was to analyze if there was a benefit to augment the washing effect before injecting BMAC in patients. 11 patients had bilateral stage I, 15 had bilateral stage II, and 22 patients had stage I on one side (10 with BMAC, 12 with WE before BMAC) and stage II on the other (12 with BMAC, 10 with WE before BMAC).

These 454 patients (267 males, 187 females) with bilateral non-traumatic osteonecrosis representing 908 hip osteonecroses were diagnosed with pre-collapse osteonecrosis (stage I or stage II), as defined by Ficat, Steinberg or ARCO classification [23]. They were 41 years old on average (range 12—pediatric patients with hematologic cancer—to 48 years) at treatment time and an average of eight years’ follow-up (a range of 5 to 10 years).

The probable etiology and risk factors among the ONFH were alcohol abuse present in 16% of patients, corticosteroid therapy in 21% of patients, sickle cell disease (SCD) in 24%, conditions such as abnormal lipid profiles (cholesterol -LDL-C, triglycerides -TGs) in 15%, hematologic or solid cancer in 10%, organ (liver, renal, cardiac) transplantation in 7%. In 8% of patients, lupus erythematosus or coagulation abnormalities, pregnancy, chronic renal failure, hyperparathyroidism, Cushing’s or Crohn’s disease was present. In the remaining 6%, the causes and factors of osteonecrosis were undetermined (idiopathic).

According to the Ficat classification [1], 434 (48%) of the 908 hips were stage I and 52% were stage II. According to the ARCO classification [1], there were 175 ARCO stage IA cases, 206 cases of ARCO stage IB, 53 cases of ARCO stage IC, 214 cases of ARCO stage IIA, 183 cases of ARCO stage IIB, and 77 cases of ARCO stage IIC at diagnosis.

### 2.2. Data Selection for Evaluation of Repair

#### 2.2.1. Tissue Source

Three different types of cells were used:

*Bone marrow concentrated cells (BMAC):* The technique has been previously reported [13,14]. Briefly summarized, bone marrow was aspirated from the iliac crest and concentrated in the cell therapy unit with GMP Facilities [13,14]. The average total number of MSCs, counted as colony-forming units-fibroblast (CFU-Fs), injected in each hip was 125,000 ± 38,000 cells (range 45,000 to 310,000 cells).

*Allogenic stem cells (Allo CT):* The transplantation was performed in specific indications (patients with cancer, patients with chemotherapy, or patients with hematologic transplantation). These patients had allogenic stem cell transplantation for their cancer from a genetically suitable matched unrelated stem cell donor. The MSCs from a matched cell donor or banked cryopreserved allogeneic MSCs derived from healthy volunteer bone marrow donors were expanded. MSCs were expanded and prepared for grafting following the standard protocols in our Good Manufacture Practice Unit, fully approved by the French Medicine Agency (FMA). Manufacture protocols are supervised and approved by the FMA for each validated advanced medicine therapy, in this case MSCs. These protocols specify the procedures for cell characterization, cell viability, cell purity, sterility tests, etc., and must follow the European Pharmacopeia.

*Autologous Expanded stem cells* (Auto Exp CT): The cell Product Manufacturing Process was obtained in GMP Facilities. An aliquot of the starting material was taken out in order to perform sterility, CFU-F test, cell count, and viability controls. Every manufacturing site followed the identical protocol outlined in ORTHO-1 CT (EudraCT 2011-005441-13) and ORTHOUNION CT (EudraCT 2015-000431-32) [22] for the entire manufacturing process. The final product used to create ORTHO-2 BM-MSCs (bone marrow mesenchymal stem cells) was the active ingredient for the quality control procedures. The active ingredient was resuspended in an albumin solution. For transportation to the operation room, 140 million cells in total—20 million cells per milliliter—were packed into one 7 mL syringe.

#### 2.2.2. Number and Characterization of Cells

*Mesenchymal stem cells (MSCs) counts*.

The analytical techniques to determine MSCs content have been reported previously [24]. Succinctly, the frequency of MSCs per 1 million nucleated cells is estimated by counting the number of colony-forming units-fibroblast (CFU-Fs) present in the culture. Cell number and proliferation were systematically assessed by DNA quantification.

Cells were further characterized by flow cytometry for surface molecular expression analysis (fluorescein isothiocyanate (FITC), phycoerythrin (PE) or allophycocyanin (APC) conjugated antibodies against CD90, CD105, CD73, or CD34). Osteogenic differentiation potential of the bone marrow (BM)-derived MSCs was evaluated by quantitative real-time RT-PCR for Runt-related transcription factor 2 (Runx2)-Hs00231692_m1; alkaline phosphatase (ALP)-Hs00758162_m1; osteocalcin (OC)-Hs00609452_g1; and bone sialoprotein 2 (IBSP)-Hs00173720_m1] when cells were cultured in classical osteogenic medium (platelet lysate (PL) was supplemented with 50 μM AA, 10 mM β-gly, and 0.1 mM Dex (Sigma Aldrich, St. Louis, MO, USA)) [25]. Scanning electron microscopy (SEM) was also used to characterize cells of patients with osteonecrosis.

We numerically compared the function of MSCs isolated from many patients with osteonecrosis to a control group of healthy donors (patients without osteonecrosis). Compared to age- and sex-matched controls, ON patients had significantly fewer BM nuclear cells and colony formation efficiency (CFE) values. On the other hand, regarding their phenotypic and functional characteristics, the MSCs from ON patients were comparable to those from the control groups. Like MSCs from healthy donors, ON-patient MSCs may be expanded for multiple passages at a similar doubling time per passage. The antigens utilized to identify BMSCs (CD105, CD90, and CD73) were expressed by cultured MSCs from both osteonecrosis and healthy donors; hematopoietic antigens were not expressed. In response to a traditional osteoinduction cocktail, these cells generated calcium deposits to a comparable degree: at 80% confluence. The controls wells (untreated: no osteoinduction) were maintained with media containing only PL. Osteoblastic gene expression revealed no appreciable distinctions between healthy and ON-MSCs, as explained in a previous paper [25].

#### 2.2.3. Liquid Injection Volume

Based on previous research [26] performed under in-vitro experimental conditions, it has been determined that the femoral head can absorb 10 cc of physiological serum without leakage. In addition, studies conducted in vivo reveal that when 20 cc is injected into the femoral head, a portion of this liquid is drained through the venous system and reintroduced into the circulation after the femoral head is washed. When core decompression was performed with fluid injection, 20 cc were injected with contrast product, demonstrating that around 10 cc remained in the femoral head. Another study was performed with radionuclide markers for some patients [13]. Most of the patients in this series treated by cell therapy received 20 cc of liquid, associating cell therapy with a washing phenomenon. When the cell therapy was an injection of cells cultured in vitro before being injected (Allo CT or Auto Exp CT), the volume of liquid was 7 cubic centimeters [22]; so, there was no washing effect and probably no passing effect in circulation or at least minimal drainage.

#### 2.2.4. Monitoring Repair with MRI

All patients enrolled in this study had a pre-operative MRI and a follow-up MRI scan five years following bone marrow cell transplantation. Due to compliance to protocols, some but not all patients had MRIs at other intervals. A 2.5 Tesla MRI scanner was utilized to acquire MRI scans, employing a 32-channel thoracic phased array coil with the subsequent pulse sequences: short TI inversion recovery sequences (TR = 5200 ms (5131–5282), TE = 50 ms (47–54), inversion time = 170 ms, FA = 111°, SL = 3 mm); T2-weighted fat saturated FSE sequence (TR = 4450 ms (2399–4450), TE = 61 ms (57–69), FA = 125°, SL = 3 mm (3–4.5)); and a flow-compensated 2D fast spoiled gradient recalled (FSPGR) sequence (TR = 21.2 ms, TE = 2.2 ms, inter-echo interval 2.2 ms, FA = 25°, SL = 3 mm). Images were analyzed using Version 6.5 of Osirix (Pixmeo SARL).

One observer (blinded to the treatment) manually delineated each of these areas and then calculated the signal-to-noise ratio (SNR) by dividing the mean signal intensity of the delineated area by the standard deviation of the background noise. For MR imaging analysis, every area was regarded as a separate observation.

The study’s second result was the volumetric assessment of the repair tissue in the osteonecrotic femoral head. Each area was considered an independent observation for MR imaging analyses.

The difference on MRI (Figure 3a,a’) between the volume before and after grafting or after core decompression was considered as a repair process. The method used in this study for the volumetric analysis of osteonecrosis with magnetic resonance imaging has been previously reported [27] and was estimated to have an accuracy of +/− two cubic centimeters. Changes below 5% of initial necrotic volume were arbitrarily considered as “no changes”, while increases or decreases of more than 5% were considered substantial. The volume percentage of the femoral head was expressed both when the femoral head was a sphere or a part of a sphere [15].

#### 2.2.5. Other Parameters

The patients’ pre-treatment conditions (cause, volume of ONFH, stage, age, genre, etc.) were reported as previously described [15,16]. The causes included the following: alcohol abuse 16%; corticosteroid use 21%; sickle cell disease 24%; abnormal lipids 15%; cancer 10%; transplantation 7%; miscellaneous 8%; undetermined 6%.

### 2.3. Statistical Analysis

We presented medians and IQR for continuous variables and frequencies and percentages for categorical variables. Logistic regression models were applied to the binary classification problems, such as the presence or absence of total repair on MRI and the presence or absence of partial repair.

The odds ratio was used to determine how each factor (stage, volume of ON, number of cells implanted, tissue source, volume of fluid injected) influenced the amount of repair. To identify the factors that have a computational association with the dependent variable, we performed a multivariate analysis of binary logistic regression (BLR) using both the forward LR and backward LR methods. The ANN used the Brain module in Stata (version 16.0) and Python (version 3.6).

Then, the patients’ pre-treatment conditions (cause, volume of ON, stage, age, genre, etc.) and the therapeutic strategies (type of cells, number of cells, volume of injection) were encoded [15,16] within the input properties using an artificial neural network [15,16] to analyze and define the importance of each factor. This allowed us to examine how well each component performed, as well as how these properties combined. While it is possible that one or more of the MSC therapy’s characteristics had no effect on ON healing on their own, their combination may allow the model to capture a significant correlation.

## 3. Results

Among the 908 hip osteonecroses, 438 received no cells, while a total of 470 hips were treated with cell therapy: seven hips received autologous expanded (Auto Exp CT) stem cells, 38 hips received allogenic stem cells, and the other 425 received BMAC.

### 3.1. Regeneration or Total Signal Regression Is Rare

The term regeneration was used when total regression of a lesion area was observed on MRI. Visual inspection of the MRI slices of all the patients with analysis of coronal, sagittal, and transverse slices demonstrated total regression of the signal for 35 hips in the whole femoral head, in the osteonecrosis area and in the decompression track canal of the osteonecrosis. This total regeneration was only observed in 7.4% of hips that received cells; total regeneration was only observed when cells were injected in a volume of 20 cc. It was never observed with decompression alone or with the washout effect; nor was it observed with allogeneic cells or with autologous expanded cells injected in a low volume (7 cc), which suggest better results with higher volume than 7 cc.

In our cohort, total regression of the signal, i.e., of the necrotic area, was only observed with autologous BMAC treatment. This does not mean that complete regeneration cannot be obtained in other patients with other cells (allogenic or expanded) or other treatment modalities; this was simply not observed in the patients included in this study.

Besides the tissue source (autologous BMAC) and the injection volume, pre-treatment conditions such as age, size of the osteonecrosis defect, its stage, and the number of cells injected played essential roles defining the treatment outcome. Only stage I osteonecrosis had a total signal regression. The other pre-treatment conditions were the patient’s age (less than 30 years), a pre-op size of osteonecrosis lower than 20 cc treated with a minimum of 150,000 cells, or some specific causes such as sickle cell disease, idiopathic, coagulation abnormality, or conditions such as abnormal lipid profiles (cholesterol -LDL-C, triglycerides -TGs), pregnancy, or idiopathic osteonecrosis.

### 3.2. Degree of Repair Analysis Using Logistic Regression in Each Group

Among the 908 hip osteonecroses of this study, 35 hips showed total regeneration. For the other 873 hips included in this study, 715 hips had some repair at the 5-year follow-up. 158 hips had no repair in any part of the femoral head despite no collapse at the 5-year follow-up.

In the following analysis, all the patients in each group had both hips treated with a different treatment on each side, as specified. Results are expressed (1) as the percentage of hips showing repair with different treatments related to the total number of hip included in this study (Figure 4); (2) as the reduction (pre- and post-treatment) in absolute volume of the osteonecrosis and as a percentage of osteonecrosis, related to the original volume of the femoral head considered as a sphere; and (3) as osteonecrosis repair volume: a difference-in-differences (D-I-D) estimation model was conducted to test differences in the osteonecrosis volume (ONV as dependent variable) before and after treatment (pre-operatively and five years post-operatively). As the volume of the femoral head is different in each patient, a “normalized” osteonecrosis repair as percentage repair of the volume of the femoral head was selected for comparison between different groups. The D-I-D estimator (function intercept) was interpreted as the mean treatment effect (normalized repair percentage) with different treatment among different groups.


**Group 1: Core decompression (CD) compared with natural evolution (NE).**


***Percentage of hips with repair:*** Among the 52 hips with core decompression, 15 hips had repair, while in the same period only six hips out of 52 with natural evolution showed some repair (15 of 52–28.8%, versus six of 52–11.5%; odds ratio 3.1081, 95% confidence interval [CI] 1.0975 to 8.8024; *p* = 0.0328).

***Reduction in volume of hip osteonecrosis**:*** For hips subjected to core decompression (CD) treatment, magnetic resonance imaging (MRI) assessment revealed a reduction in the average pre-operative osteonecrosis volume from 17.2 cc (with a range of 11–31 cc) to 12.1 cc (with a range of 6–21 cc) at the latest follow-up. In terms of percentage of the femoral head volume, the initial necrotic tissue volume (see Figure 5a) decreased from 28.4% of the original femoral head volume (standard deviation 15.2%) to 17.1% (SD 13.2%). For hips with natural evolution (NE), MRI demonstrated a decrease in osteonecrosis volume from an average 16.4 cc (range 10–32 cc) to 13.3 cc (range 8–30 cc) at the most recent follow-up; as a percentage of the volume of the femoral head, the decrease (Figure 3a) moved from 26.8% (SD 16.3%) to 21.2% (SD 14.4%).

***Osteonecrosis repair volume:*** With CD treatment, the absolute mean repair volume was 5.1 cubic centimeters (range 0 to 10 cc) and was significantly (*p* = 0.04) higher than with natural evolution (average 3.1 cc; range 0 to 5 cc). Based on the D-I-D model, the mean “normalized osteonecrosis repair” percentage was 11.3% of the femoral head volume in the CD treatment, compared to 5.6% in the NE.

**Group 2: the Washing Effect (WE) improved the repair of CD**.

***Percentage of hips with repair:*** Among the 50 hips with CD, only 13 hips had repaired, while over the same period of 5 years, 19 hips had some repair among those 50 when WE was applied (26% versus 38%; odds ratio 0. 0.5322, 95% confidence interval [CI] 0.2148 to 1.3183; *p* = 0.1729).

***Reduction in volume of hip osteonecrosis:*** For hips with CD, MRI demonstrated a decrease in the pre-operative average volume from 21.4 cc (range 13–35 cc) to 13.5 cc (range 7–25 cc) at the most recent follow-up. As a percentage of the volume of the femoral head, the decrease (Figure 5b) moved from 23.3% (SD 14.6%) to 18.3% (SD 12.9%). For hips treated with WE, MRI evaluated a reduction in the pre-operative average volume of ONFH from 23.7 cc (with a range of 11–32 cc) to 11.1 cc (with a range of 6–22 cc) at the 5-year follow-up. Expressed as a percentage of the femoral head volume, a reduction of 12.2% of the necrosis was observed (from 25.4% (SD 13.8%) to 13.2% (SD 10.9%).


**Group 3 (BMAC versus CD): Bone marrow concentrated cells increased the repair.**


***Percentage of hips with repair:*** A total of 234 hips were treated with BMAC and had an incidence of repair of 71.8% at 5 years. In comparison, the incidence of repair was 26% at 5 years among the 234 contra-lateral hips treated with CD without cells (71.8%, 168 among 234 versus 15%, 61 among 234; odds ratio 7.2191, 95% confidence interval [CI] 4.8018 to 10.8533; *p* < 0.0001).

***Reduction in volume of hip osteonecrosis:*** For the 234 hips without injection of cells (CD) the pre-operative average volume of ONFH decreased from 19.2 cc (range 12–38 cc) to 12.4 cc (range 6–24 cc) at the most recent follow-up; as a percentage of the total volume of the femoral head (Figure 5c), a decrease from 27.4% (SD 12.9%) to 20.1% (SD 10.8%) was quantified.

For the 234 hips treated with cells (BMAC), the pre-operative average volume decreased from an average 23.7 cc (range 15–35 cc) to 7.8 cc (range 0–13 cc) at the most recent follow-up; as a percentage of the volume of the femoral head, the decrease (Figure 5c) moved from 29.8% (SD 14.2%) to 12% (SD 9.3%).

***Rate of reduction in volume of osteonecrosis with BMAC:*** In Group 3, due to the high number of patients and to different follow-up protocols, 125 patients had MRI at more frequent intervals during the 5-year follow-up, allowing the analysis of the reduction in percentage of the osteonecrosis volume as a function of time between 3 months and 5 years after the implantation of BMAC (Figure 6). Among those 125 hips with BMAC and several MRIs, the percentage of the volume of the femoral head necrosis moved from 29.8% (SD 14.2%) pre-operatively to 22.1% (SD 16.3%) at one-year follow-up (FU), 15.3% (SD 15.4%) at 2-year FU, 13.1% (SD 13.1%) at 3-year FU, and to 12% (SD 9.3%) at 5-year follow-up. This corresponds to a reduction of almost half of the original ONFH volume in the 1st year and further indicates that most of the defect’s regeneration occurs during the first 2 years post-surgery.


**Group 4 (BMAC versus WE):**


***Percentage of hips with repair:*** A total of 25 hips were treated with BMAC and had an incidence of repair of 68% (17 among 25) at five years. In comparison, the incidence of repair was lower (*p* = 0.012) with 32% (eight among 25) at five years in hips treated with washing effect.

***Reduction in volume of hip osteonecrosis:*** For the 25 hips with WE, MRI showed a decrease in the pre-operative average volume from 20.3 cc (range 12–38 cc) to 11.8 cc (range 6–24 cc) at the most recent follow-up; as a percentage of the volume of the femoral head, the decrease (Figure 5d) moved from 26.1% (SD 15.2%) to 19.1% (SD 12.7%). For the 25 hips treated with BMAC, the pre-operative average volume of ONFH decreased from an average 25.2 cc (range 14–32 cc) to 10.3 cc (range 0–15 cc) at the 5-year follow-up; as a percentage of the volume of the femoral head, the decrease (Figure 5d) moved from 30.2% (SD 15.7%) to 11.2% (SD 7.2%), thus, a 36% recovery.

***Osteonecrosis repair volume:*** For the 25 hips treated with BMAC, the mean volume of repair evaluated by MRI at the most recent follow-up was 14.9 cubic centimeters (range 0 to 24 cubic centimeters) and significantly (*p* = 0.031) higher than for the 25 hips treated with WE (8.5 cubic centimeters, minimum 0, maximum 12 cc). Based on the D-I-D model, the mean “normalized osteonecrosis repair” percentage was 19% in the BMAC treatment and only 7% in the WE treatment.

**Other Groups 5, 6, 7**.


**
*Percentage of hips with repair:*
**


No significant difference was found between autologous BMAC, allogenic (Allo CT), and autologous expanded cells (Auto Exp CT) treatments (Figure 4). Repair occurred in a similar percentage of hips independently from the tissue source, respectively, 71%, 72%, 75%, and 73% for BMAC, Allo CT, Auto Exp CT, BMAC + WE, despite a much higher number of cells with expanded allogenic or autologous cells (>100 million in both expanded cells groups) versus 100,000–200,000 cells with BMAC).

***Reduction in volume of hip osteonecrosis***: *In Group 5 (BMAC versus Allo CT)*, the repair displayed a similar pattern on both sides: with BMAC, a mean pre-operative ONV of 28% (SD 12.2%) as a percentage of the femoral head, dropping to 9.2% (SD 7.6%), compared with a drop from 25.3% (SD 14.4%) to 9.3% (SD 4.9%) on the side with allogenic cells (*p* = 0.32, both paired *t*-test).

*In Group 6 (BMAC versus Auto Exp CT)*, the mean percentage of osteonecrosis volume dropped from pre-operative 25.2% (SD 13.4%) to 7.2% (SD 5.3%) on the side with BMAC and dropped from 24.3% (SD 14.7%) to 8.8% (SD 4.9%) on the side with expanded cells (*p* = 0.21, both paired *t*-test). Despite a higher number of cells, the volume of repair was lower (but not significantly) with expanded cells, and the comparison does not allow any conclusive quantitative analysis. *In Group 7*, *a pre-operative washing of 20 cc improves the repair with BMAC injection*; there was a small difference in the volume of repair between hips with cell therapy and washing effect and those without washing with 20 cc, but the difference was not significant; they, respectively, drop from pre-operative 28.2% (SD 13.4%) to 10.2% (SD 6.2%) on the side with BMAC and drop from 25.3% (SD 14.7%) to 7.8% (SD 4.9%) on the side with expanded cells (*p* = 0.21, both paired *t*-test).

***Osteonecrosis repair volume:*** *For Groups 5*,*6*,*7* a D-I-D estimation model was conducted to test differences in normalized osteonecrosis repair percentage among the different groups. D-I-D estimator (function intercept) was interpreted as the mean treatment effect (normalized osteonecrosis repair percentage) on BMAC cases compared to allogenic expanded cells or autologous expanded cells. D-I-D estimations were not conclusive (Figure 5e–g) in favor of any tissue source group. This may be due to the low number of hips in each group, suggesting that the mean reduction in ON-V was similar in the three groups (*p* = 0.453, all paired *t*-test).

### 3.3. Treatment Effects Analysis Comparison between Groups

When grouping all the hips according to the treatments they received, 352 hips were treated with BMAC, 336 hips had only core decompression, and 75 hips had a washing effect treatment. Fifty-two hips were followed with natural evolution, while 38 received allogenic cells, seven received auto expanded cells, and 48 others were treated with BMAC associated to a washing effect. Based on the D-I-D model, the mean highest “normalized osteonecrosis repair” was 18.8% and observed with Allo CT in 38 hips of pediatric patients. Normalized osteonecrosis repair is a percentage of a sphere adjusted to the volume of the femoral head, which allows comparison with other treatments. On average for the 352 hips treated with BMAC, the “normalized osteonecrosis repair” was 18.1% (range 0% to 26%), while it was 15.5% for the seven hips with Auto Exp CT.

In the absence of cells for the 336 hips treated with core decompression alone, the “normalized osteonecrosis repair” was 8.2% (range 0% to 13%).

As a remark, the average volume of the hip osteonecrosis included in this series was 20.2 cc (range 10 −35 cc) for a mean volume of the femoral head of 65.5 cc (range 38.8 to 113.1) calculated as a sphere. So, the “normalized osteonecrosis volume” was 30.8% (range 11% to 39%).

### 3.4. Influence of Other Parameters on Cell Based Repair

Figure 7 shows an AI evaluation of the possible influence of different parameters on ONFH repair. The patients’ pre-treatment conditions (cause, volume of ON, stage, age, genre, etc.) and the therapeutic strategies (type of cells, number of cells, volume of injection) were encoded within the input properties for the model to make analysis and predictions. We first trained a neural network to take only one input parameter and predicted the osteonecrosis repair score.

The full set of parameters was then provided as inputs to further train the neural network. A correlation test was first performed between all parameters to make sure that no pairs of input parameters were closely correlated. Both Pearson’s Correlation and Spearman’s Rank Correlation coefficients were proven to be smaller than 0.50. The parameter identified as most correlated to repair was the osteonecrosis volume percentage, with an R2 value of 0.74, followed by stage (0.71), cause (0.67), and age (0.62). Whereas parameters related to the treatment strategy, such as the implantation cell number, the tissue source, and washing effect impacted the outcome to a lesser extent (with, respectively, R2 of 0.59; 0.55, and 0.53). These input parameters were studied in descending order of importance.

**Stage and pre-op volume percentage**.

We first compared the two most important properties: stage of ONFH and pre-op volume percentage.

This study showed that there is a “critical size” of ON that cannot effectively repair by itself in the absence of cells even in stage I ON but also that critical thresholds of the osteonecrosis volume exist for effective repair to happen even in presence of cells implantation (Table 1). No repair was obtained with CD (alone or with washing) when the pre-op ON volume was higher than 20% of the calculated femoral head volume in stage I osteonecrosis, and no repair was obtained in stage II osteonecrosis with CD (alone or with washing). When patients received cells, regardless of cell type, for hips with stage I, a repair process could be observed for necrosis with pre-op volume ON percentage of up to 40%; while for hips with stage II, the repair process only occurred for a pre-op volume of less than 30%.

When patients received cells, the percentage of repair was greater for stages I than for stages II. When considering the 190 stage-I hips that received BMAC, these hips received an average number of 117,000 ± 42,000 cells (range 45,000 to 301,000 cells), leading to a decrease in the osteonecrosis size (as a percentage of the volume of the femoral head) from 31.4% (SD 16.5%) to 10.4% (SD 7.2%). When combining all the 216 stage-II hips that received BMAC, these 216 hips received a higher (*p* = 0.031) number of 127,000 ± 39,000 cells (range 48,000 to 310,000 cells), but the decrease in size of osteonecrosis was significantly less (*p* = 0.021), moving from 28.4% (SD 15.3%) to 19.3% (SD 13.4%).

**Cause of osteonecrosis**.

Using AI and machine learning, for the same stage, the same pre-op volume of osteonecrosis and the same number of implanted cells, some diseases or comorbidities showed better repair, the best repair being obtained in sickle cell disease. According to the data obtained in vitro on the quality of cells, this variation is not related to any impaired functions of the cells in some diseases. This may rather be attributed to the different bone qualities and different cytokines presence in bone for each cause of osteonecrosis. To grade the repair related to each cause, the causes were graded with odds ratios. The causes are ranked in Table 2 in decreasing order of the repair effect.

**Age of the patient**.

There was a negative correlation between the age of the patient (Figure 8) and the number of MSCs that were present in the iliac crest (r = −0.42;). However, this decrease was not the same in all the pathologies. While not very significant in sickle cell disease, (probably because all the patients are relatively young), the decrease in the number of MSCs with age was quicker in patients with some conditions such as corticosteroids intake and cancer. This was a cause of variation in the number of cells implanted in the case of BMAC treatments (and Auto Exp CT).

Independently of the number of MSCs implanted, patients younger than 30 years had a significantly better volume of repair when compared with those older than 30 years, and this was more significant for stage II than for stage I lesions (Table 3).

**Implanted cell number**.

Figure 9 shows a low increase in the normalized percentage of osteonecrosis repair when the number of implanted cells increased from 10,000 to 100,000. Above 100,000 implanted cells and up to 1,000,000, a significant positive but low correlation was observed between the number of MSCs in the graft and the volumes of repair obtained at the latest follow-up, regardless of the cell source (r= 0.31; *p* = 0.001). Further increases in cell number did not show any further beneficial effect on the percentage of ON healing.

**Tissue source**.

The tissue sources of MSCs, concentrated bone marrow (BMAC), allogenic cells, and autologous expanded cells were used as a parameter in our model. These three sources of MSCs are the most widely used and studied. The number of patients treated with allogenic and expanded cells was low in our database (Figure 2). Data on BMAC were abundant. The machine learning results suggested that cell-based treatment in general (BMAC, allogenic, and expanded MSCs) benefits healing outcomes, but no individual tissue sources (BMAC, allogenic or, expanded MSCs) were shown to be more beneficial than the others.

**Washing effect**.

From this machine learning model, the least important parameter appeared to be the washing effect when cells are injected. There was a difference between the core decompression alone and core decompression with washing effect; some differences in the repair volume were also observed when hips had a washing effect of 20 cc before receiving 20 cc volume of BMAC. The washing effect was more efficient for stage II hips and large-volume osteonecrosis. But, due to the small number of patients available in that group, no significant difference was observed, despite the washing effect suggesting a benefit.

## 4. Discussion

### 4.1. Osteonecrosis Repair: Evolutionary Vestige with a Very Low Spontaneous Capacity

Regeneration is a process triggered by injury involving a combination of synergistic processes. Its purpose is not only to limit the damage caused by the injury but also to generate new cells that can replace the lost tissue. Several animal species [28,29] are known for their remarkable regenerative abilities, including fully regenerating limbs. As an example, axolotls, amphibians native to Mexico, can fully regrow a functional limb within a few months after amputation. Similarly, zebrafish can regenerate various body parts, including their fins.

Mammals, including humans, have limited regenerative abilities compared to certain other animals. There are a few examples of limited regenerative capabilities in mammals. Male deer [28], such as the red deer or the white-tailed deer, can regenerate their antlers, which, after shedding, regrow from the base, guided by a specialized tissue called the velvet, a rich source of nutrients supporting their growth. Interestingly, bone and limb regenerative capacity differs among species and between genetically identical organisms at different life stages. Neonatal mice have been shown to exhibit some regenerative abilities in certain tissues during the early postnatal period. For instance, neonatal mice [30] can regenerate the tips of their digits (fingertips) if amputated within the first few days after birth, including the regrowth of skin, soft tissues, bone, and even nail structures.

Osteonecrosis is considered one of the planet’s oldest known diseases, with evidence recovered from extended reptiles (continuing diving), 250 million years ago [31]. The ichthyopterygians were marine reptiles that appeared 240 million years ago and became extinct around 90 million years ago. In the Late Triassic, they developed a fish- or dolphin-like body shape, and 16–20% of their humeri have revealed signs of bone necrosis. Comparisons with living reptiles and other marine animals provide insights into the regenerative potential of ichthyosaurs. Therefore, osteonecrosis regeneration has probably been lost during evolution, even though the ability to repair fractures has been conserved throughout evolution in the human species.

### 4.2. Why Is Osteonecrosis Repair More Limited Than Fracture Healing in the Same Bone?

The osteonecrosis process also leaves extensive necrotic cell and fat debris in the marrow space, requiring macrophage clearance through phagocytosis [17,18,19]. The necrotic microenvironment promotes a chronic inflammatory response marked by increased migration, activation, and persistence of macrophages in the repair tissue with increased inflammatory cytokine production. While acute inflammation is required for bone repair, chronic inflammation leads to further tissue damage and fibrosis. Unlike fractures, where all the debris and dead cells are easily cleaned by the adjacent structures of the soft tissues, osteonecrosis of the femoral head occurs in a closed system where there is no possibility that this debris can escape outside the bone. Therefore, unlike fracture repair, where acute inflammation quickly subsides in coordination with activation of bone regenerative processes, bone repair following ischemic osteonecrosis is marked by a chronic inflammatory process that may impair bone regeneration.

At the time when the corresponding author introduced the technique (1985) of injecting fluid in the osteonecrosis, the idea was not to wash the osteonecrosis but rather to inject a contrast material to study the venous return of the osteonecrosis and at the same time inject a diluted anesthetic solution into the femoral head to improve post-operative analgesia. The retrospective review of the MRIs showed a better osteonecrosis repair when the contrast product and anesthetic solution was injected. The injected fluid was around 20 cc; the rationale for 20 cc was to obtain a dilution of the contrast material and the anesthetic product. Experimental studies carried out later [26] for other reasons showed that the absorption capacity of a human femoral head is limited to 10 cc and that any additional volume of fluid injected will be taken up by the nearby venous drainage. The fact that some patients had better results when we performed this 20 cc injection technique associated with drilling suggests that washing the debris located in the femoral head has an effect. This washing effect may decrease necrotic cell debris and inflammatory molecules that negatively affect bone repair. This is in accordance with research performed in piglets [32], in which bone-formation was increased with washing in induced osteonecrosis.

### 4.3. Background of Cell Therapy

The traditional treatment for early-stage ONFH is core decompression [11], which can release pressure and open the small vessels blocked by pressure in the femoral head, thus relieving pain. However, imaging data from post-operative follow-ups have shown that the necrotic area of the femoral head did not shrink significantly in many patients, eventually leading to the collapse and deformation of the femoral head [10].

Therefore, core decompression alone cannot satisfactorily reconstruct and repair the femoral head necrosis area. In 1999, Hernigou [33] reported a decrease in the activity number of stem cells in ONFH [24] when evaluated on bone marrow aspirates. This decreased activity number may affect local tissue and vascular regeneration, oxygen supply, and osteogenic function, eventually leading to ONFH. The concept of performing stem cell transplantation for treating early-stage ONFH was first proposed by Hernigou in 1987 and reported in the English-language literature in 2002 [13]. At that time, there were two reasons to inject a concentrated volume of bone marrow of 20 cc into the femoral head. On the one hand, previous experience of drilling without cells showed that only the fact of injecting a solution of 20 cc of contrast product and anesthetic solution seemed to improve the repair; on the other hand, due to the fact that the bone marrow concentration technique was limited, it was necessary to inject a sufficient volume to have a significant number of stem cells.

A recent study evaluating the results of this technique by artificial intelligence [15] showed that the factors were highly numerous. Therefore, the current study aimed to investigate the influence of the factors that appeared to be the most important for the repair of osteonecrosis in the absence of collapse, which can, of course, modify the action of these factors.

Because the focus of this article is on the effect of cells on the regeneration of ONFH, we further focused on the effect of cells treatment and target-related factors as input properties, including the tissue source (autologous, allogenic, expanded); cell number and volume of injection; and stage, volume, and cause of osteonecrosis. Since the latter (stage, cause, and volume) can modify the treatment outcome, selecting patients with bilateral osteonecrosis who received different treatments on each side was important to avoid certain biases.

### 4.4. Therapeutic Response Disparity: Therapy Puzzle

Since the repair is much better when there are cells than when there are none, we can consider that there is a minimum required number of cells to obtain repair.

However, according to our data, there is probably no ideal number of MSCs (even with millions of cells) that guarantee total repair or regeneration in osteonecrosis, and the critical number of cells allowing for a partial repair could depend on many factors, including the cause of osteonecrosis. We can only speculate on a minimum threshold value from a theoretical point of view and from the observation of patients.

*Theoretically*, a femoral head with a volume of 50 cc will contain approximately 35,000 MSCs [26]. This value can thus be considered as the target number of MSCs to be implanted in an osteonecrotic femoral head to achieve the same number of MSCs present before osteonecrosis. Now, if a patient has an osteonecrotic lesion of 17 cc and a femoral head of 50 cc (thus the lesion is roughly 1/3 the head volume), the minimum number of cells to inject is approximately 12,000 MSCs (i.e., 35,000/3). However, as mentioned above, there is a reduction in the number of MSCs in the area surrounding the osteonecrosis, so the number of MSCs to be injected in the above cited example should exceed 12,000. It is also necessary to consider that only a portion of bone marrow cells or expanded cells will remain in the femoral head after implantation due to venous drainage. Homing studies have been performed by Hernigou and Beaujean [13] and Gangji [34] to determine this proportion. Both groups showed that between 30% and 50% of the injected pre-labeled cells remained in the femoral head 24 h after implantation. This proportion may vary according to the volume injected in the femoral head. Indeed, a larger volume may increase the proportion of cells lost by venous drainage, while a smaller volume might not fill the entire volume of the osteonecrotic lesion. So, from a theoretical point of view, the minimum number of MSCs to inject should be greater than 24,000 MSCs and is probably situated between 24,000 and 35,000.

*From a clinical point of view*, MRI shows that an increased number of cells up to 1,000,000 leads to an increased volume of repair in osteonecrosis, indicating a direct correlation between cell number and volume of repair. Beyond this dose, the repair reaches a plateau or declines a little. When we exceed a million cells, the curve bends slightly negatively. The determination of MSC dose for therapy remains intuitive in current clinical practice. A wide range of implanted cell numbers has been found in the literature, ranging from a few thousand to several million [35,36]. Besides the implantation number, cells were transplanted at various densities in different clinical trials, ranging from a thousand to a million cells per milliliter of the delivery agents.

In the treatment, there is a simultaneous incorporation of factual elements and those derived from intuition. When osteonecrosis is treated with cell therapy, the number of cells that are injected is known. However, the density of cells per milliliter is a product of intuitive reasoning. This derivation hinges on variables such as the initial product, injection volume, osteonecrosis volume, and the femoral head volume, where diffusion occurs either passively through fluid pressure or actively via chemotaxis related to cytokine concentration [37,38,39,40]. Consequently, these elements can amalgamate with therapeutic effects in a manner exhibiting sub-additive, super-additive, or modular relationships. For instance, an exploration into the interactive combination of items such as cell number, concentration, injection volume, osteonecrosis volume, and femoral head volume reveals specific potential therapeutic effects [41,42,43,44,45]. Assuming the acceptance of each item having a relationship with a therapeutic effect, it is crucial to recognize that the relationship between cell number, concentration, and injection volume may undergo a substitutive change at the time of injection. Post-injection, this relationship undergoes a transformation contingent on the diffusion of the product within the femoral head. Considering the femoral head as a closed space, injecting the same cell quantity in a small volume poses the risk of exclusively filling the osteonecrosis, resulting in no washing effect within the femoral head. Conversely, injecting the same cell quantity in a large volume may extend beyond the femoral head with some link outside the femoral head but achieve full filling of the femoral head with a washing effect. Consequently, the relationship may exhibit super-additive characteristics until a certain volume threshold, after which it becomes sub-additive. This association likely exhibits a modular effect. The combination may be the surgeon’s choice or a pure hazard if the osteonecrosis is not correctly targeted and cells are injected outside the osteonecrosis.

The mechanisms of necrosis repair through MSC therapy may share many commonalities, such as differentiation of the MSCs at the damage site, secretion of regenerative factors, immune regulation, and decreasing the amount of necrotic cells, debris and inflammatory molecules abounding in the femoral head [46,47,48,49,50]. We were surprised to have the same repair with the million MSCs expended the in vitro and bone marrow aspirate that contains fewer MSCs and CFUs for an equal volume. But one should keep in mind that these cells were pure MSCs without macrophages or any cell type present in bone marrow and were injected in low volume. There were, therefore, no cells allowed to resorb the debris. Also, no washing effect was involved, due to the low volume. Conversely, when the concentrated bone marrow is injected, it contains the cells, some macrophages, and some polynuclear cells. This “soup” undoubtedly allows the easy resorption of debris and molecules and may contain other factors/cells promoting bone healing. In addition, BMAC is injected in a volume of 20 cc, which has a mechanical washing effect.

Even if, whatever the origin of the cells, the effect seems the same in the different groups, the choice of the type of cells must be discussed in some circumstances. Using autologous concentrated cell therapy (BMAC) in solid or hematologic cancer patients is a therapeutic challenge; a low level of MSCs is observed in this population due to previous treatment with chemotherapy. Intervention employing in-vitro-expanded autologous MSCs could be discussed but would necessitate a two-step approach: first bone-marrow harvesting, followed by a surgical intervention 3 to 4 weeks later. For patients with hematologic cancers, the situation is different when they have been treated with allogenic bone marrow transplantation [51]. They are chimers; using autologous MSCs in chimers patients may also be challenging, particularly if they have a ghost-versus-host disease (GVH). As an alternative strategy, the use of their donor-banked cryopreserved allogeneic MSCs could be considered as a better possibility for those patients who had received allogeneic bone marrow transplantation and afterwards have a GVH disease. Without GVH, MSCs in the iliac crest remain of host origin even after allogeneic bone-marrow transplantation. Therefore, both allogenic or autologous host MSC treatment approaches are possible.

### 4.5. Poor versus Robust Regenerators Patients

Beyond optimizing stem cell products, the potential value of candidate recipient selection has to be considered using distinct strategies, as some patients with some diseases answer better and others have less repair for the same ONFH stage, the same volume, and the same number of cells.

Conventionally, the selection of candidates for ON stem cell therapy relies on a cut-off value, such as ON size, and no collapse [52,53,54,55,56,57]. However super responders, as in sickle cell disease, defined as achieving a better repair, could be treated when osteonecrosis is with a higher size and in later stages (stage II), as compared with poor or non-responders; adjustment of therapeutic goals should match the individual patient [58,59,60,61,62,63].

The factual parameters are stage, pre-op volume, and the presence of some diseases. But many parameters that are usually reported as a cause and associated factors derive from our biological intuitions [64,65,66,67] and are not factual. The best examples are probably associated factors such as alcohol abuse, smoking, or corticosteroid medication. The value of these items may interact and combine, namely alcohol abuse, smoking, lipid profiles, and corticosteroid medication. When the patient both receives corticoids and has a history of alcohol abuse, the question is, when both factors are present, how the risk changes. Is there a sub-additive relationship where the whole is less than the sum of the parts or a super-additive relationship where the whole is more than the sum of the parts? The problem is the same when the patient has an abnormal lipid profile and receives corticosteroids. This event, of course, is factual and it is derived from our intuition that the consequence for repair could not be 1 + 1 = 2, but less than 1 or more than 2. And the relation is complex, even when analyzed with artificial intelligence. Now, taking into consideration smoking and alcohol, smokers may not need as much alcohol and vice versa. They may be mutually redundant and decrease each other’s value, since once you have had a glass of wine, a cigarette is less necessary and less desirable. Thus, this could be a sub-additive relationship, where the whole is less than the sum of the parts. A few of the items may not affect so much each other’s risk for repair with cell therapy, as with some modular relationships.

With aging, the proliferative and functional abilities of MSCs are impaired because of a combination of intrinsic and environmental factors. As proper bone healing requires an inflammatory phase, the increased survival of anti-inflammatory M2 macrophages and reduced secretion of pro-inflammatory factors with age may jeopardize timely bone regeneration. At the same time, aging negatively impacts MSC proliferation and differentiation, further impeding the bone-healing process. It would appear that, taken together, both macrophages and MSCs, cells critical for regeneration of musculoskeletal tissues, are adversely affected by aging. This scenario provides new opportunities for modulation of cellular events to optimize the healing of mesenchymal derived tissues, including bone. It is therefore necessary to consider the patient’s age in the number of cells to be aspirated in the iliac crest as well in the therapeutic indications.

### 4.6. Limitation of This Study

Our study has some limitations. The main limitation of this study is the reduced sample size in some groups, while other groups had a larger number of patients. Consequently, some comparisons were surely underpowered. However, due to a bilateral comparison of hips in the same patients with the same cause of osteonecrosis, many biases are avoided in this study. The second limitation of this study is that it only concerns the reduction in volume on the MRI and that it does not concern the histological aspect of the repair in the hips which had no collapse and, therefore, no possibility of carrying out a histological study on the anatomical parts. Another study is underway to correlate MRI aspects and histological repair aspects of osteonecrosis. Another significant limitation of this study is that it does not address the prevention of the risk of collapse, unlike previously published research [15]; despite a repair process observed in the femoral head, collapse is possible if the percentage of the femoral head repaired is not sufficient or if the location of the osteonecrosis is critical.

## 5. Conclusions

This study examines the effectiveness of mesenchymal stem cell (MSC) therapy in treating human hip osteonecrosis, encompassing a significant sample of 908 cases. The research effectively delineates the influence of various treatment modalities, including core decompression and different MSC therapies (autologous bone marrow concentrate, allogeneic expanded, and autologous expanded cells) on the outcome of hip osteonecrosis repair. The results underscore a crucial correlation between the number of MSCs applied and the extent of osteonecrosis repair, noting an optimal threshold beyond which additional MSCs do not yield proportional benefits. The stage and the volume of osteonecrosis and the count of MSCs are critical factors that dictate the success of the repair process. Furthermore, this analysis provides valuable insights into the limitations and potentials of MSC therapy in clinical settings, emphasizing the necessity for tailored therapeutic approaches based on individual patient characteristics such as age, disease as a cause of osteonecrosis, and specific hip osteonecrosis conditions (stage and volume). The detailed exploration of MSC therapy paves the way for more refined, effective, and patient-specific strategies in managing hip osteonecrosis.

## Figures and Tables

**Figure 1 cells-13-00776-f001:**
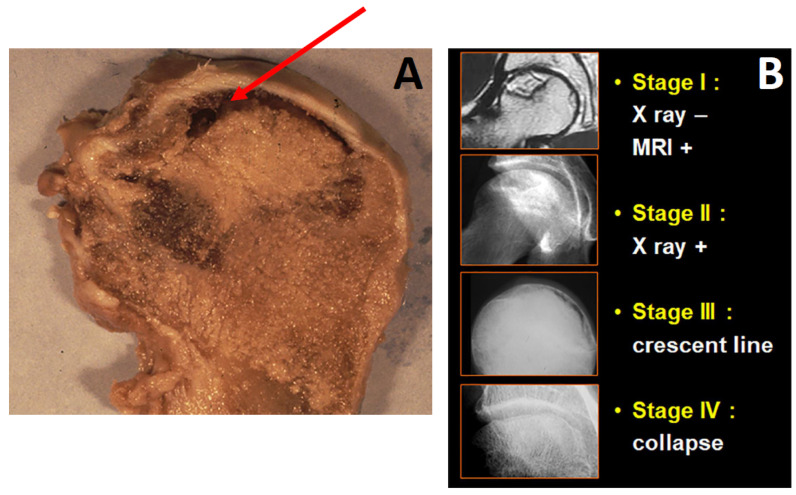
(**A**): Femoral head osteonecrosis: the arrow shows the loss of sphericity of the femoral head and the collapse. (**B**): Classification of hip osteonecrosis: At stage I (osteonecrosis visible on MRI) and stage II (osteonecrosis visible on radiographs), the femoral head is spherical. At stage III and stage IV, there is a deformation of the femoral head.

**Figure 2 cells-13-00776-f002:**
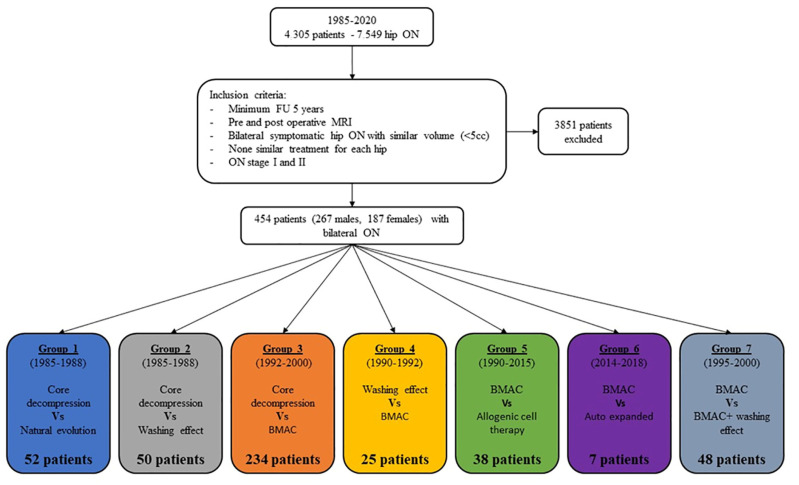
Inclusion and repartition of treatments in the different groups.

**Figure 3 cells-13-00776-f003:**
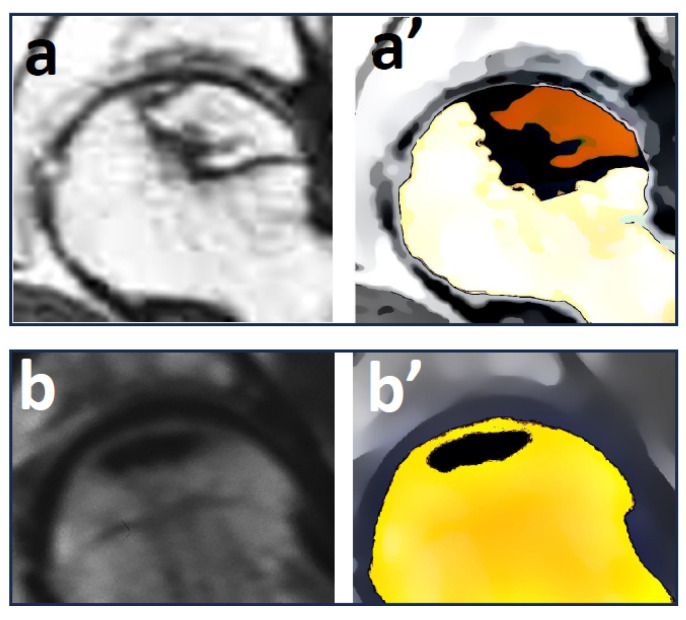
(**a**,**a’**): Volume of osteonecrosis on pre-operative MRI. (**b**,**b’**): Volume of hip osteonecrosis after repair.

**Figure 4 cells-13-00776-f004:**
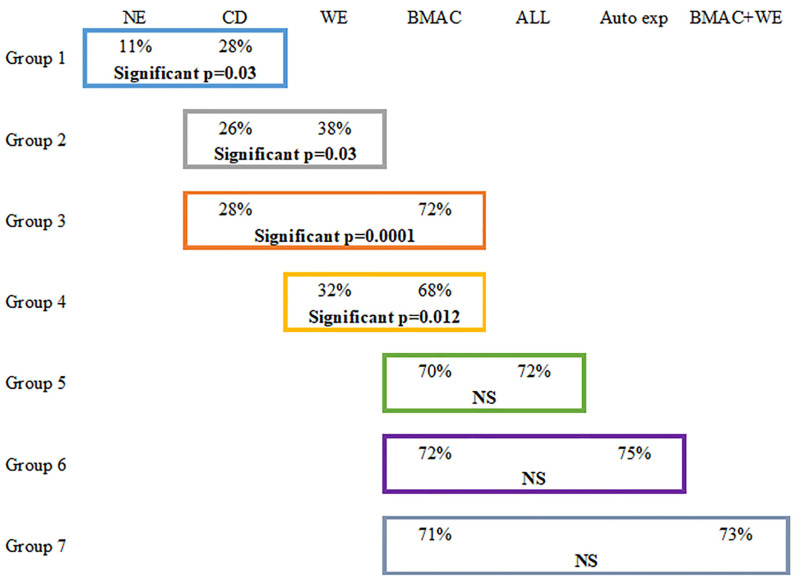
Percentage of hip repair in each group; NE natural evolution; CD: core decompression; WE: washing effect; BMAC: bone marrow autologous concentration; ALL: allogenic stem cells; autologous expansion in vitro.

**Figure 5 cells-13-00776-f005:**
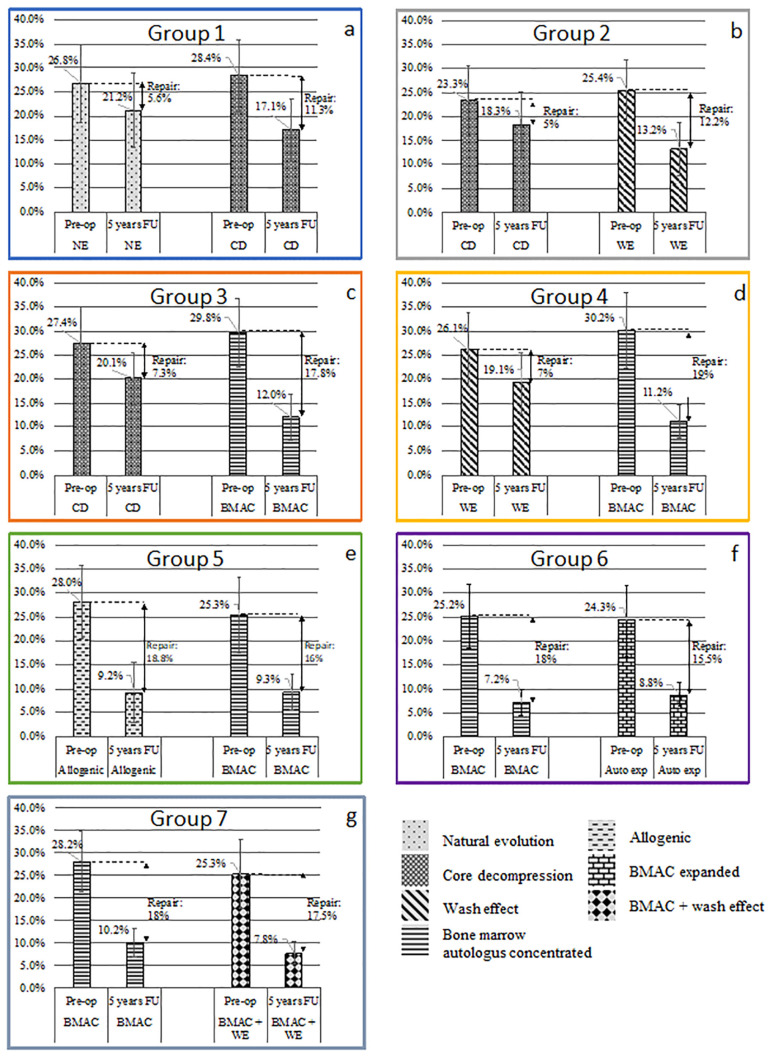
Reduction in volume of hip osteonecrosis in each group. (**a**–**g**) represents Group 1–7.

**Figure 6 cells-13-00776-f006:**
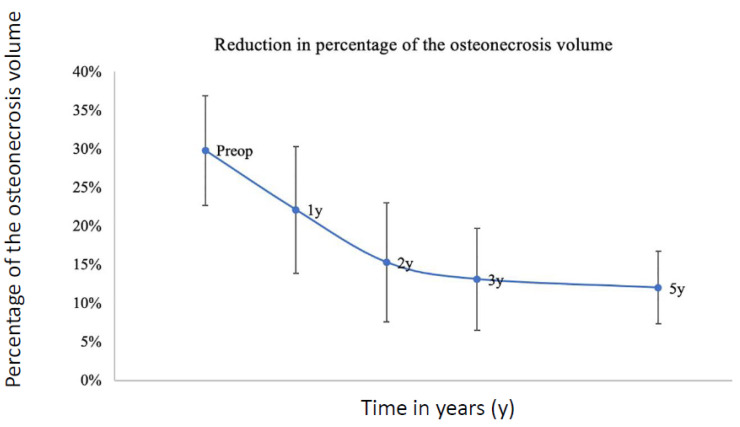
Rate of reduction in volume of osteonecrosis in Group 3 from pre-operative to 5-year follow-up. *Y* axis: percentage of the osteonecrosis axis. *X* axis: time in years.

**Figure 7 cells-13-00776-f007:**
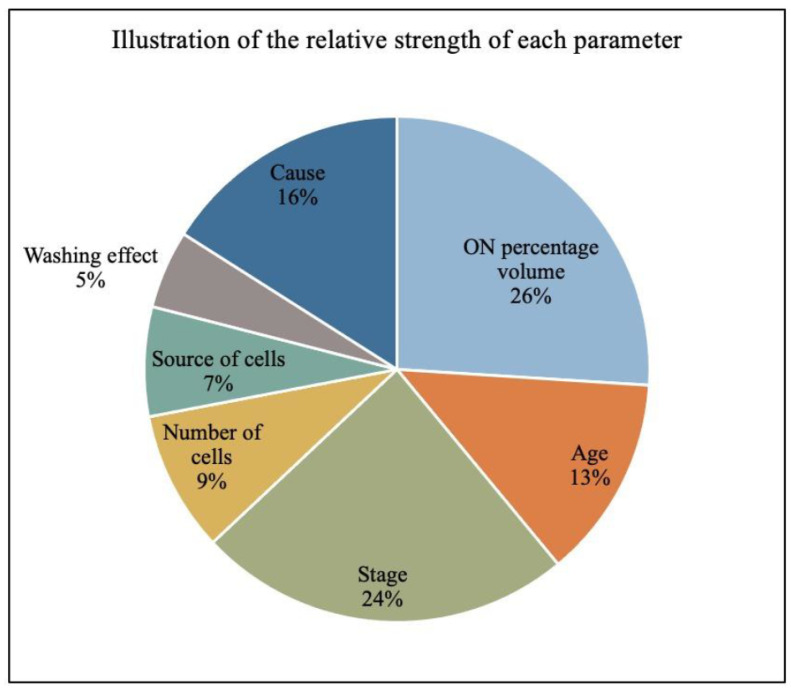
Evaluation of the possible influence of different parameters on osteonecrosis repair.

**Figure 8 cells-13-00776-f008:**
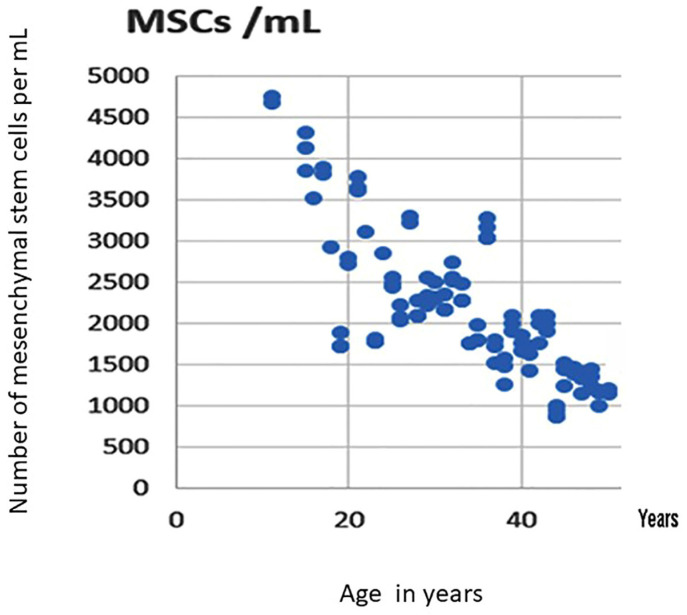
Correlation between the age of the patient and the number of MSCs that were present in the iliac crest. *X* axis: age in years. *Y* axis: number of mesenchymal stem cells per mL.

**Figure 9 cells-13-00776-f009:**
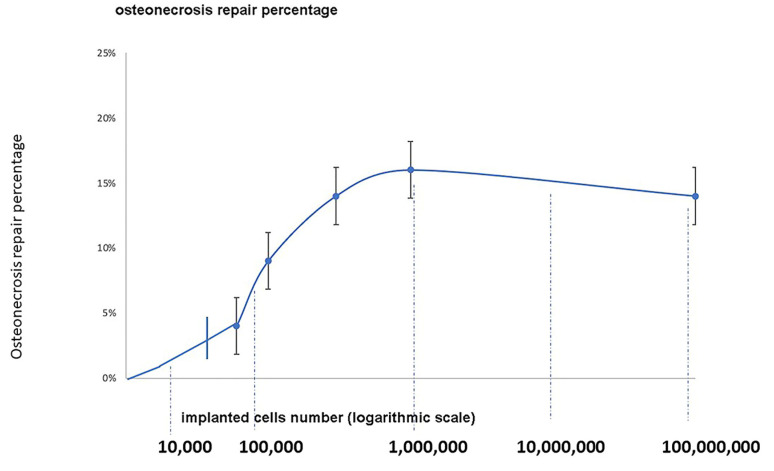
Increase in the normalized percentage of osteonecrosis repair when the number of implanted cell number increased up to one million. *Y* axis: osteonecrosis repair percentage. *X* axis: implanted cells number.

**Table 1 cells-13-00776-t001:** Critical threshold value of volume percentage to observe repair.

		Eventual Repair	No Repair
No cell	Stage I	≤20%	>20%
	Stage II	Never repair	
Cells	Stage I	≤40%	>40%
	Stage II	≤30%	>30%

**Table 2 cells-13-00776-t002:** Cause of osteonecrosis and effect on repair.

*Blood Disorders*		*Metabolic Diseases*		*Corticosteroids for*	
sickle cell disease		Pregnancy	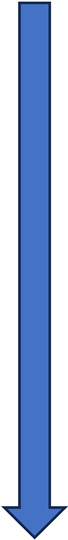	lupus erythematosus	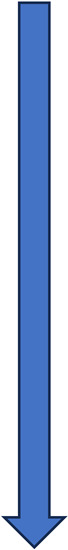
thalassemia	Gaucher disease	uveitis
polycythemia	Gout	multiple sclerosis
hemophilia	HIV infection	Crohn’s disease
lupus erythematosus	Pancreas	pemphigus vulgaris
coagulation abnormalities	Hemochromatosis	nephrotic syndrome
cancer, such as leukemia	chronic renal failure	liver transplantation
		hyperparathyroidism	renal transplantation
** *Associated factors:* **		Cushing’s disease	aplastic anemia
«Caisson disease»	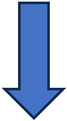	MTHFR	cardiac transplantation
alcohol abuse				
smoking				

**Table 3 cells-13-00776-t003:** Reduction in the percentage of the volume of ON according to age, stage, and initial volume of ON among hips treated with cells.

Status at ⊄ Implantation	Reduction in % of Volume ON		
	Age ≤ 30 Range (12;30)	Age > 30 Range (31;48)	*p*
Stage I	32% ± 10%	31% ± 15%	<0.01
Stage II	30% ± 16%	29% ± 20%	<0.01
ON volume >20%	38% ± 21%	37% ± 29%	<0.01
ON volume ≤20%	15% ± 5%	16% ± 10%	<0.01

## Data Availability

Data supporting reported results can be found at the Henri Mondor Hospital.

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
