# Peer review of "Mesenchymal Stem Cell Therapy for Bone Repair of Human Hip Osteonecrosis with Bilateral Match-Control Evaluation: Impact of Tissue Source, Cell Count, Disease Stage, and Volume Size on 908 Hips"

_cells, 2024, doi:10.3390/cells13090776_

Round 1
Reviewer 1 Report
Comments and Suggestions for Authors
The authors of the article titled “Mesenchymal Stem Cell Therapy for Bone Repair on Human Hip Osteonecrosis with Bilateral Match-Control Evaluation: Impact of Tissue Source, Cell Count, Disease Stage, and Volume Size on 908 hips.” investigated the efficacy of MSC therapy in treating human hip osteonecrosis. They assessed how various factors, such as tissue source, cell count, and disease stage, affect treatment outcomes. They aimed to identify optimal treatment protocols to enhance the efficacy of MSC therapy in bone repair. This study used a single-center retrospective approach to analyze the outcomes of various MSC therapies and core decompression across 908 cases. The methodology provides a good basis for comparing treatment effects, which, in my opinion, was reinforced by appropriately chosen and used statistical analyses such as logistic regression and multivariate analysis, which were reinforced by machine learning methods. The data are presented in tables and figures, facilitating an understanding of the complex matter of the described topic and the interaction between many variables affecting the treatment.
The manuscript is well structured and effectively communicates its findings to an audience that is more likely to be comprised of medical professionals and less for researchers. Even brief notes on the analytical methods used and not only rough citations would be of use in this case. This article progresses logically from the introduction of the research problem to a detailed analysis and discussion of the results. And I must admit that the narrative flow of the “story” in the introduction and the discussion sections are very pleasing for the reader. The quality of writing was high, with clear explanations of clinical and scientific data, making complex information accessible and understandable. References are appropriately cited with a focus on recent and relevant literature that supports the study's claims.
The conclusions drawn are well supported by the data and highlight the potential of tailored MSC therapy to improve the treatment outcomes for hip osteonecrosis. This study acknowledges its limitations, including its retrospective design and single-center scope, which might affect the generalizability of the results. Nevertheless, the implications for future research and clinical practice are significant, suggesting that further studies should explore these findings in a prospective or multicenter format to enhance their robustness.
Overall, this manuscript makes a valuable contribution to the fields of regenerative medicine and orthopedics. This suggests that modifying MSC therapy parameters can significantly affect treatment efficacy, thereby providing a basis for further research and potential clinical applications. This paper is suitable for publication in Cells, with minor revisions.
Minor comments:
1. In the Introduction, there is no clearly stated hypothesis under the study. For an article investigating multiple variables affecting mesenchymal stem cell therapy outcomes in hip osteonecrosis, a hypothesis could specifically state the expected relationships or predictions based on previous research or theoretical considerations. For example, the authors hypothesized that higher cell counts and specific tissue sources would result in significantly improved bone repair outcomes in early stage hip osteonecrosis compared to lower cell counts and different tissue sources.
2. Table 1 – it looks more like figure and should be consider like this, and should have figure captions included.
3. r=0.31 is a low correlation and should be described as this, although it is significant.
Comments on the Quality of English Language1. Lines 664-666
Should not ti be like this?
This is in accordance with research performed in piglets [32] in which bone formation was increased with washing in induced osteonecrosis.
Author Response
comment 1: agree; added in the manuscript: lines 114 to 116
comment 2: it is a table with figures captation
comment 3: agree added in the manuscript line 584
Agree for line 664-666: correction done in the text
Reviewer 2 Report
Comments and Suggestions for Authors
The manuscript is a very helpful asset to the discussion about stem cell therapy here in hip necrosis. It’s very well orchestrated and scientifically sound in a critical way, that’s rare in a world of a still ongoing stem cell hype. There are no specific comments, sure some groups are underpowered, but the overall conclusions are worthwhile.
Author Response
thank you